# Learning to Correct Noisy Labels for Fine-Grained Entity Typing via Co-Prediction Prompt Tuning

**Minghao Tang**[1,2], **Yongquan He**[3], **Yongxiu Xu**[1,2]*, **Hongbo Xu**[1], **Wenyuan Zhang**[1,2]
and   **Yang Lin**[3]

[1]Institute of Information Engineering, CAS, China
[2]School of Cyber Security, UCAS, China
[3]Meituan, China
{tangminghao,xuyongxiu,hbxu}@iie.ac.cn, heyongquan@meituan.com

## Abstract

Fine-grained entity typing (FET) is an essential task in natural language processing that aims to assign semantic types to entities in text. However, FET poses a major challenge known as the noise labeling problem, whereby current methods rely on estimating noise distribution to identify noisy labels but are confused by diverse noise distribution deviation. To address this limitation, we introduce Co-Prediction Prompt Tuning for noise correction in FET, which leverages multiple prediction results to identify and correct noisy labels. Specifically, we integrate prediction results to recall labeled labels and utilize a differentiated margin to identify inaccurate labels. Moreover, we design an optimization objective concerning divergent co-predictions during fine-tuning, ensuring that the model captures sufficient information and maintains robustness in noise identification. Experimental results on three widely-used FET datasets demonstrate that our noise correction approach significantly enhances the quality of various types of training samples, including those annotated using distant supervision, Chat-GPT, and crowdsourcing.

## 1 Introduction

Fine-grained entity typing (FET) is a fundamental task in the field of natural language processing (NLP). It involves assigning specific types to entity mentions based on the contextual information surrounding them. The results of FET can be leveraged to enhance various downstream NLP tasks, including entity linking  (Raiman and Raiman, 2018; Chen et al., 2020a), relation extraction (Shang et al., 2020, 2022), question answering (Wei et al., 2016) and other tasks (Jiang et al., 2020; Liu et al., 2021a).

FET poses a formidable challenge that entails not only the classification of entities across diverse domains but also the distinction of entities that

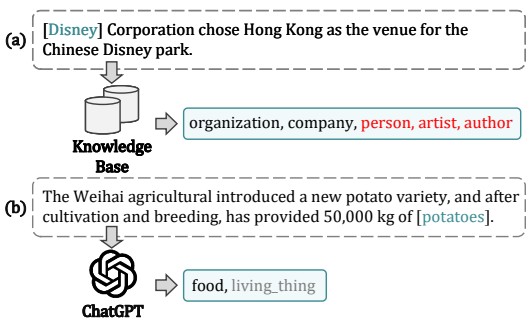

Figure 1: Examples from OntoNotes dataset, with inaccurate labels in red and unlabeled labels in gray. (a) A distantly labeled example. (b) A weakly labeled example by ChatGPT.

may exhibit superficial resemblances but possess distinct underlying meanings. For example, the term "apple" could pertain to a technology corporation or a fruit. Achieving high performance in FET typically requires a considerable amount of annotated data, a resource that is both costly and time-consuming to acquire. To tackle this issue, researchers have explored various approaches for automatically labeling training samples. One particularly popular method is distant supervision (Ling and Weld, 2012; Gillick et al., 2014), which assumes that any text mentions an entity in existing knowledge bases (e.g., Freebase (Bollacker et al., 2008)) is related to the corresponding entity types. However, this technique can be susceptible to noise and ambiguity (refer to Figure 1a), since it fails to consider the contextual information surrounding the entities.

Meanwhile, the large language models (LLMs), such as ChatGPT[1], has drawn significant interest among researchers in exploring their applications in text annotation tasks. Recent studies (Gilardi et al., 2023; Törnberg, 2023) indicate that zero-shot ChatGPT classifications outperform crowd-workers. However, despite their remarkable suc-

---

*Yongxiu Xu is the corresponding author

[1]https://openai.com/blog/chatgpt

cess in other domains, the weakly labels ChatGPT generated for FET still pose challenges. As shown in Figure 1b, it can generate accurate weakly labels, but its coverage of fine-grained types may be limited. This difficulty may arise from the challenge of designing a prompt that can accurately handle fine-grained entity types across multiple domains simultaneously. However, this incomplete recognition also results in noisy labels, as it mislabels the correct type as a negative sample.

Training deep models on data with noisy labels can result in the learning of incorrect patterns and subsequent incorrect predictions. As a result, several approaches have been proposed to tackle the issue of noisy labels in FET, including designing robust loss functions (Ren et al., 2016; Xu and Barbosa, 2018), estimating noise transition matrices (Wu et al., 2021; Pang et al., 2022), and correcting noisy labels (Onoe and Durrett, 2019; Zhang et al., 2021; Pan et al., 2022). Among these approaches, correcting noisy labels is often considered the most desirable, as it leads to more accurate and informative labels. However, the methods for noise correction often rely on estimating the distribution of noise, which can be challenging due to the diverse deviations arising from different labeling methods. For instance, distant supervision tends to generate inaccurate labels, while labels generated by ChatGPT are accurate but incomplete.

In this paper, we propose a novel noise correction method for FET using the co-prediction prompt-tuning technique, which leverages multiple prediction results to identify and correct noisy labels. Specifically, we first fine-tunes a pre-trained masked language model (PLM) on data with noisy labels using a co-prediction prompt. This prompt contains two mask tokens, which focus on different predictive capabilities and produce multiple predictions for each entity mention. Note that the presence of noisy labels may make it difficult for the two masks to arrive at a consensus on the outputs. In this scenario, one mask may start fitting the noise before another, leading to divergent co-predictions. Hence, we design an optimization objective concerning divergent co-predictions during fine-tuning to maintain the fitting difference of masks on the noise labels and robustness to noise identification. Afterward, we integrate the multiple prediction results to recall unlabeled labels and use a differentiated margin to identify inaccurate labels.

We conduct experiments on three public FET datasets: OntoNotes and WiKi have distantly annotated training data; Ultra-Fine has crowdsourced training data. The experimental results show that the performance of a baseline model is significantly improves after training on the corrected data, indicating the effectiveness of our noise correction method. Additionally, we utilized ChatGPT to relabel a subset of training data from OntoNotes and Wiki and then optimized this data using our method. The experimental results demonstrate that our method can further improve the quality of the data annotated by ChatGPT.

## 2 Related Work

### 2.1 Automatic Labeling Techniques for FET

Due to the scarcity of manually labeled data, automatic labeling techniques have gained significant attention in FET. Some studies (Ling and Weld, 2012; Gillick et al., 2014) utilize distantly supervised learning to generate labeled training samples by mapping entity mentions to entities from public knowledge bases and selecting reliable mapping types as labels. However, the labels obtained through this method are independent of sentence semantics. On the other hand, some studies (Choi et al., 2018; Dai et al., 2021) use head supervision to label training samples, where head words are used to label entities. However, both distant supervision and head supervision have limitations in producing accurate labels.

Recently, large language models such as Chat-GPT have shown promising results in weakly supervised learning, even surpassing crowdsourced annotators in some domains (Gilardi et al., 2023; Törnberg, 2023). However, due to the large number of entity types, designing effective prompts for ChatGPT to generate comprehensive label sets can be challenging in FET. In this paper, our goal is to develop a noise correction method that enhances the quality of weakly labeled data, instead of designing perfect prompts for ChatGPT or rules for distantly supervised learning.

### 2.2 Noise Learning Method for FET

The problem of noisy labeling presents a significant challenge in FET research. Several research studies have explored label dependencies. Ren et al. (2016) proposes a hierarchical partial-label loss to handle noisy labels. Wu et al. (2019) leverages a random walk process on hierarchical labels to weight out

noisy labels. Wu et al. (2021) learns the noise confusion matrix by modeling the hierarchical label structure. However, these approaches require a predefined hierarchical label structure, which can be difficult to establish in practice.

Some studies assume that a subset of clean labels is available. Onoe and Durrett (2019) trains noise filter and relabel modules on the clean data. Pang et al. (2022) treats the clean labels as anchors to estimate noise transition matrices. However, the efficacy of these methods is heavily reliant on the size of the clean labels, which may not always be guaranteed. Hence, some approaches try to estimate noise distribution to produce pseudo-truth labels (Zhang et al., 2021) or filter out noisy labels (Pan et al., 2022). Nonetheless, unifying the noise distribution from different automatic labeling methods can be challenging. Differ from them, our approach employs co-prediction prompt tuning for noise correction, thus avoiding the pitfalls of estimating noise distribution for various types of weakly labels.

## 3  Methodology

In this section, we introduce the problem formulation and describe our noise correction method. As shown in Figure 2, our method mainly consists of two main steps: 1) Applying a co-prediction prompt to fine-tune PLMs with weakly labeled corpora; 2) Correcting noisy labels by analyzing the co-prediction results of this model.

### 3.1  Problem Formulation

The FET task aims to assign appropriate semantic types to an entity mention $m$ based on contextual information provided by a sentence $x$. The semantic types are a subset of a pre-defined set of fine-grained entity types $Y = \{y_1, y_2, \ldots, y_t\}$. If the training corpora is labeled by distant or weak supervision, two types of noisy labels may arise. The first type is inaccurate labeling, which occurs when an annotator assigns an incorrect semantic type to an entity mention. The second type is unlabeled labeling, which happens when the annotator fails to identify an appropriate semantic type.

In our approach, the goal is to improve the quality of the training corpus by correcting noise labels, specifically by removing inaccurate labels and recalling unlabeled labels.

### 3.2  Co-Prediction Prompt Tuning

The accurate identification of inaccurate or unlabeled labels is a crucial challenge in noise correction, given that manual identification of such labels is time-consuming and expensive. Recent studies (Han et al., 2018; Nguyen et al., 2020) have highlighted the "memory effect" (Arpit et al., 2017), which suggests that deep models tend to memorize clean labels before fitting to noisy labels when trained on data with noise. Building on this insight, we propose leveraging a co-prediction prompt-tuning technique for noisy label detection that explicitly captures the fitting difference.

### 3.2.1  Model Structure

There are two primary components in the prompt-based model: prompt and verbalizer. The prompt provides task-related information to guide the pre-trained language model generating relevant and accurate output. The verbalizer converts the output of the model into natural language.

**Co-Prediction Prompt**  Differ from traditional prompts, we adopt a co-prediction prompt for FET that contains two different mask tokens: [PMASK] and [NMASK]:

$$\mathrm{T}(\mathbf{c}, \mathbf{m}) = \{\mathbf{c}\} \, [\mathbf{P}] \, \{\mathbf{m}\} \text{ belongs to} \\ \text{[PMASK] rather than [NMASK],} \quad (1)$$

where $[\mathbf{P}]$ represents a set of soft words that are randomly initialized trainable special tokens, and [PMASK] and [NMASK] are initialized by the embedding of original mask token [MASK]. Clearly, artificially words (e.g., "belongs to", "rather than") are used to guide the two mask tokens focus on different predictive abilities. The objective is to extract diverse knowledge from PLMs and create varying difficulty in fine-tuning the representations of the two mask tokens.

**Verbalizer Selection**  Following the previous prompt-based model (Ding et al., 2021), we utilize the soft verbalizer that stores a set of mappings from a soft word $v \subseteq V_y$ to a label $y \subseteq Y$. As fine-grained entity types typically have hierarchical type structure, such as "/organization/company/news", we use the average embedding of type tokens to initialize soft words.

To determine the probability distribution of entity mentions $m$ with respect to the label set $Y$, the co-prediction model generates two distinct scores for each entity type $y \in Y$ by using [PMASK] and

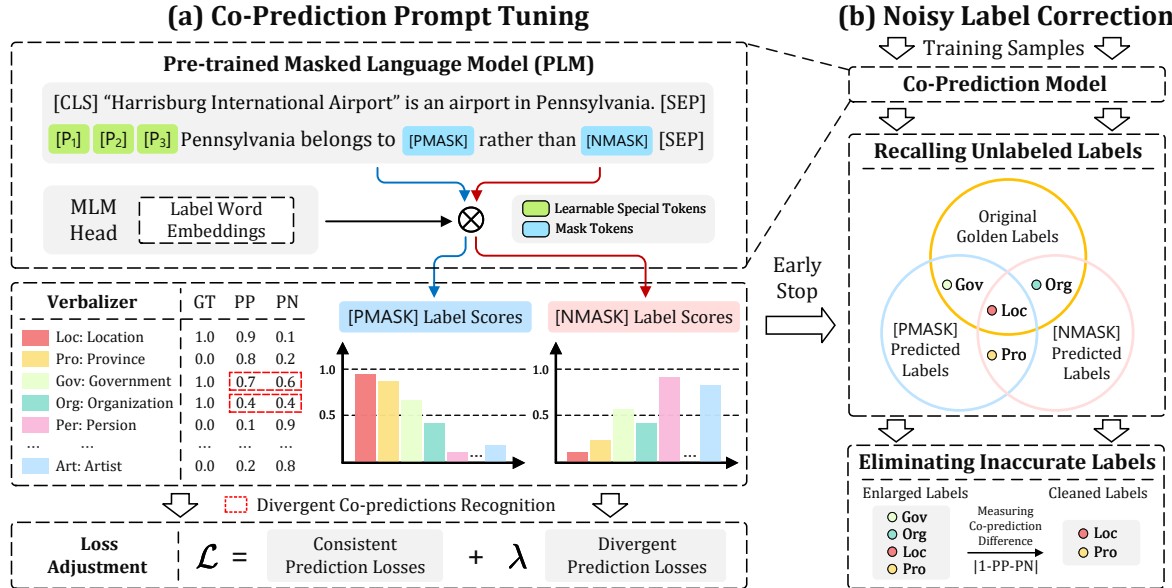

**(a) Co-Prediction Prompt Tuning**

**(b) Noisy Label Correction**

Figure 2: Overview of our noise correction method, which first fine-tunes a pre-trained masked language model via co-prediction prompt-tuning and then use this model for noise correction. "GT" denotes the ground-truth. "PP" and "PN" are prediction scores of [PMASK] and [NMASK].

[NMASK]. The prediction score for each label $y$ can be calculated as follows:

$$p_{p,y}|(\mathbf{c}, \mathbf{m}) = p([\text{PMASK}] = v|\text{T}(\mathbf{c}, \mathbf{m})), \quad (2)$$

$$p_{n,y}|(\mathbf{c}, \mathbf{m}) = p([\text{NMASK}] = v|\text{T}(\mathbf{c}, \mathbf{m})), \quad (3)$$

where $p_{p,y}$ and $p_{n,y}$ are the prediction scores of [PMASK] and [NMASK] for label $y$.

### 3.2.2 Training with Noise

Drawing on insights from "memory effect" (Arpit et al., 2017), it is possible for one mask to fit noisy labels more quickly than another due to differences in their fine-tuning speeds. This fitting discrepancy can result in divergent predictions on noisy labels. Figure 2a provides an example in which the predictions generated by [PMASK] and [NMASK] for entity types "Gov" and "Org" are inconsistent. In accordance with the semantics of the co-prediction prompt, we define **divergent co-predictions** that satisfy one of the following criteria:

$$\begin{cases} p_{p,y} \geq 0.5 \ and \ p_{n,y} \geq 0.5 \\ p_{p,y} < 0.5 \ and \ p_{n,y} < 0.5 \end{cases} \quad (4)$$

**Optimization Loss Function**   As the fine-tuning process progresses, both [PMASK] and [NMASK] will eventually fit all noisy labels, leading to a decrease of divergent co-predictions. To maintain the ability of co-prediction model to generate divergent results on noisy labels, we propose constraining the

model's learning from the labels that exhibit divergent co-predictions. Hence, we adjust the training loss function as follows:

$$L(\mathbf{m}, \mathbf{c}) = \gamma \sum_{Y_k} L_{y,k} + \sum_{Y_{t-k}} L_{y,t-k}, \quad (5)$$

where $Y_k$ denotes the labels with divergent co-predictions, $k$ is the number of these labels; $Y_{t-k}$ denotes the labels with consistent co-prediction, $t$ is the total number of labels; $L_y$ is the co-prediction loss for each label $y \in Y$; $\gamma$ is a hyper-parameter that represents the loss weight.

As [PMASK] and [NMASK] focus on opposite learning abilities, the co-prediction loss $L_y$ is calculated as follows:

$$\begin{aligned} L_y = \ &\text{BCE}(p_{p,y}|(\mathbf{m}, \mathbf{c}), \ \hat{p}_y) + \\ &\text{BCE}(p_{n,y}|(\mathbf{m}, \mathbf{c}), \ 1 - \hat{p}_y), \end{aligned} \quad (6)$$

where $\hat{p}_y \in \{0, 1\}$ denotes the ground-truth for whether the entity mention $m$ should be classified as label $y$ given the context $c$, and $\text{BCE}(\cdot)$ denotes the binary cross-entropy loss function.

During the initial training stage, it is crucial to recognize that labels with divergent co-predictions may include a large number of clean labels. Therefore, we start the model training with $\gamma = 1$. As the fine-tuning process progresses, we gradually decrease $\gamma$ until it reaches the marginal value. Once the co-prediction model achieves the peak generalization performance, we terminate the model training process. We evaluate the effectiveness of this

training strategy through empirical experiments in Section § 5.3.

### 3.3 Noisy Label Correction

In this section, we will detail how to correct noisy labels in the training data by recalling unlabeled labels and eliminating inaccurate labels.

**Recalling Unlabeled Labels** The co-prediction model utilizes two mask strategies, [PMASK] and [NMASK], to extract knowledge from PLMs. By combining the predictions generated by both strategies, we can produce a more comprehensive set of predicted labels, some of which might be unlabeled in the training data. As shown in Figure 2b, the predicted label "Pro" is not included in the original golden labels, which suggests that it may be a potentially unlabeled label.

**Eliminating Inaccurate Labels** After obtaining a more comprehensive label set by recalling unlabeled labels, the next step is to optimize this set by removing inaccurate labels. As the co-prediction model retains the ability to generate divergent predictions on noisy labels, we can identify inaccurate labels by measuring the absolute difference in the co-prediction scores. Specifically, we can calculate the divergence score $\delta_y$ for each label $y$ as follows:

$$\delta_y = |p_{p,y} - (1 - p_{n,y})| \qquad (7)$$

We can then classify each label $y$ as either a clean or inaccurate label by comparing its divergence score $\delta_y$ to a margin threshold $\epsilon$. If $\delta_y > \epsilon$, we classify the label $y$ as inaccurate and remove it from the label set.

Different weak supervision methods may result in different levels of label noise, so it is important to choose an appropriate margin threshold $\epsilon$ for each method. For example, distant supervision often results in inaccurate labels, so it may be more appropriate to use a small $\epsilon$ to remove such noise. In contrast, ChatGPT may produce accurate but incomplete labels, in which case a relatively larger $\epsilon$ could be adopted to ensure that more potentially relevant labels are retained.

## 4 Experimental Settings

### 4.1 Datasets

We evaluate our noise correction method on three publicly FET datasets. **OntoNotes** (Weischedel et al., 2013) used sentences from OntoNotes corpus, which was distantly annotated using DBpedia

| Datasets | WiKi | OntoNotes | Ultra-Fine |
|---|---|---|---|
| hierarchy depth | 2 | 3 | 1 |
| entity types | 112 | 89 | 2519 |
| mentions-train | 2009898 | 253241 | 2000 |
| mentions-dev | - | 2202 | 2000 |
| mentions-test | 563 | 8963 | 2000 |

Table 1: Statistics of datasets.

Spotlight (Gillick et al., 2014), with 253K training samples and 89 entity types. **WiKi** (Ling and Weld, 2012) used sentences from Wikipedia articles and news reports, and was distantly annotated using Freebase (Bollacker et al., 2008), with 2M training samples and 112 entity types. **Ultra-Fine** Choi et al. (2018) collected 6K samples through crowd-sourcing, which contains 2,519 entity types. Statistics of datasets are in Table 1.

Moreover, we employ ChatGPT to relabel 3k and 6k training samples from OntoNotes and WiKi, respectively. For Ultra-Fine, it is challenging to design a comprehensive prompt for data relabeling with a huge number of entity types. More details are in Appendix A.

### 4.2 Baseline Methods

We first consider previous noise relabeling or correction methods for FET, including LDET (Onoe and Durrett, 2019), NFETC-AR (Zhang et al., 2021), and ANLC (Pan et al., 2022). Meanwhile, we also compare our method with some noise learning methods for FET, including AFET (Ren et al., 2016), NFETC (Xu and Barbosa, 2018), and FCLC (Pang et al., 2022). Finally, we compare our method with other competitive FET systems, such as UFET (Choi et al., 2018), ML-L2R (Chen et al., 2020b), MLMET (Dai et al., 2021), LRN (Liu et al., 2021b), UNIST (Huang et al., 2022), and PKL (Li et al., 2023).

### 4.3 Experimental Details

In our co-prediction model, we choose the BERT-base (Devlin et al., 2019) model as backbone. Then, the parameters of BERT are optimized by Adam optimizer (Kingma and Ba, 2015) with the learning rate of 2e-6, 2e-6 and 2e-5 on WiKi and OntoNotes, and Ultra-Fine, respectively. Other hyperparameters are in Appendix B.

After obtaining the corrected training sets of each dataset, we train a basic prompt-based model using supervised learning to evaluate the effective-

| Model | Acc | Macro F1 | Micro F1 |
|---|---|---|---|
| **With Distantly Annotated Training Samples** | | | |
| AFET(2016) | 55.3 | 71.2 | 64.6 |
| NFETC$_{hier}$(2018) | 60.2±0.2 | 76.4±0.1 | 70.2±0.2 |
| ML-L2R(2020b) | 58.7 | 73.0 | 68.1 |
| LRN(2021b) | 56.6 | 77.6 | 71.8 |
| NFETC-AR$_{hier}$(2021)[†] | 64.0±0.3 | 78.8±0.3 | 73.0±0.3 |
| FCLC$_{hier}$(2022) | **65.3±0.2** | 79.6±0.3 | 74.0±0.3 |
| DenoiseFET(2022)[†] | 59.2±0.2 | 81.3±0.3 | 75.3±0.4 |
| **Baseline** | 54.8±0.7 | 78.1±1.2 | 71.2±0.9 |
| **Baseline** (corrected) | 58.8±0.8 | 83.7±0.2 | 76.3±0.3 |
| w/ co-predict | 57.0±0.5 | 83.1±0.1 | 75.2±0.2 |
| **With ChatGPT Annotated Training Samples** | | | |
| **Baseline** | 64.4±0.7 | 85.8±0.3 | 80.5±0.5 |
| **Baseline** (corrected) | 64.6±0.7 | **87.1±0.1** | **81.7±0.1** |
| w/ co-predict | 64.3±0.6 | 86.5±0.2 | 81.3±0.1 |

Table 2: Performance on OntoNotes test set. [†] denotes the previous noise relabeling or correction approach.

| Model | Acc | Macro F1 | Micro F1 |
|---|---|---|---|
| **With Distantly Annotated Training Samples** | | | |
| AFET(2016) | 53.3 | 69.3 | 66.4 |
| NFETC$_{hier}$(2018) | 68.9±0.6 | 81.9±0.7 | 79.0±0.7 |
| ML-L2R(2020b) | 69.1 | 82.6 | 80.8 |
| NFETC-AR$_{hier}$(2021)[†] | 70.1±0.9 | 83.2±0.7 | 80.1±0.6 |
| FCLC$_{hier}$(2022) | **71.3±1.1** | 82.2±0.7 | **81.1±0.6** |
| DenoiseFET(2022)[†*] | 67.5±0.3 | 82.3±0.6 | 78.5±0.6 |
| **Baseline** | 58.9±0.5 | 81.1±0.2 | 76.7±0.2 |
| **Baseline** (corrected) | 66.4±0.2 | **84.3±0.2** | 80.1±0.2 |
| w/ co-predict | 66.9±0.4 | 84.2±0.1 | 79.9±0.1 |
| **With ChatGPT Annotated Training Samples** | | | |
| **Baseline** | 62.2±0.6 | 80.8±0.1 | 77.5±0.3 |
| **Baseline** (corrected) | 65.8±0.8 | 82.6±0.4 | 79.1±0.4 |
| w/ co-predict | 65.1±0.7 | 82.4±0.3 | 78.6±0.4 |

Table 3: Performance on WiKi test sets. [†] denotes the previous noise relabeling or correction approach. [*] means our re-implementation via their code.

ness of our approach[2]. For this baseline, we fine-tune the BERT-base model with a standard prompt (without *"rather than* [NMASK]*"*).

## 4.4 Evaluation

Following prior works, we use the strict accuracy (Acc), Macro F1, and Micro F1 scores for OntoNotes and WiKi. As for Ultra-Fine, we use the Macro precision (P), recall (R), and F1 scores. Moreover, we conduct our experiment five times and report the mean and standard deviation values.

---

[2]Our code link: https://github.com/mhtang1995/CPPT

---

| Model | P | R | F1 |
|---|---|---|---|
| **With Crowdsourced Annotated Training Samples** | | | |
| UFET(2018) | 47.1 | 24.2 | 32.0 |
| LDET(2019)[†] | 51.5 | 33.0 | 40.2 |
| LRN(2021b) | 54.5 | 38.9 | 45.4 |
| MLMET(2021) | 53.6 | 45.3 | 49.1 |
| UNIST(2022) | 49.2 | **49.4** | 49.3 |
| DenoiseFET(2022)[†] | 55.6±0.4 | 44.7±0.3 | 49.5±0.1 |
| PKL(2023) | 52.7 | 49.2 | **50.9** |
| **Baseline** | **57.0±0.7** | 41.2±0.3 | 48.0±0.2 |
| **Baseline** (corrected) | 53.2±0.7 | 48.1±0.5 | 50.5±0.1 |
| w/ co-predict | 52.7±0.6 | 48.2±0.5 | 50.3±0.1 |

Table 4: Performance on Ultra-Fine test set. [†] denotes the previous noise relabeling or correction approach.

## 5 Results and Discussion

### 5.1 Main Results

**Results on Distantly Annotated Data** Table 2 and 3 present the results on OntoNotes and WiKi with distantly annotated training set. Compared with previous SOTA methods for FET, the simple baseline (BERT-base + Prompt Tuning) achieves on-par or inferior performance when training on the original training set. Afterward, this baseline model achieves the best Macro F1 scores over all datasets when training on the corrected training set, outperforming previous SOTA methods by a magnitude from 1.1 to 2.4 percentages.

**Results on ChatGPT Annotated Data** Tables 2 and 3 also present the results on the OntoNotes and Wiki using a ChatGPT annotated training set. It is noteworthy that the baseline model achieves comparable or superior performance with a limited number of ChatGPT annotated training samples, as compared to a significantly larger number of distantly annotated training samples (3k vs. 253k in OntoNotes, and 9k vs. 2M in Wiki). Furthermore, we apply our noise correction approach on these training samples, resulting in significant performance improvements of the baseline model across all metrics. These results provide further evidence of the effectiveness of our approach in accurately identifying and correcting noisy labels.

**Results on Crowdsourced Annotated Data** Table 4 presents the results on Ultra-Fine. The performance of the baseline model is significantly improved after training on the corrected training set, achieving comparable results with the previous best method. This finding indicates that our

noise correction method can enhance the quality of human-annotated training data, even in a challenging dataset such as Ultra-Fine.

Furthermore, compared with previous best noise correction method (DenoiseFET), our approach has the advantage of being straightforward to implement, which does not require counting the noise distribution of each type and training additional models for noise correction.

## 5.2 Ablation Study

To evaluate the contribution of each component in our noise correction method, we conduct ablation studies on the three datasets.

As the co-prediction prompt plays a crucial role for noise correction, we first assessed its impact on the model's performance. To be specific, the case *"w/ co-predict"* denotes a baseline model (BERT-base + Co-Prediction Prompt Tuning) are trained on the corrected training set. Performance of [PMASK] are reported in Table 2, 3 and 4. However, the results show a slight decline in performance when the co-prediction prompt are used, which suggests that it may not enhance the peak generalization performance of the model.

Next, we investigated the impact of each step in the noise correction process § 3.3. As shown in Table 5, the case *"w/o unl."* indicates that the noisy labels are corrected without recalling unlabeled labels. As we can see, the baseline model achieves higher precision but lower recall scores over the all datasets. These results demonstrate the effectiveness of this step in recalling unlabeled labels and can be applied to various common types of weak labels. The case *"w/o ina."* indicates that the noisy labels are corrected without eliminating inaccurate labels. Similarly, as shown in Tables 5, the baseline model achieves relatively lower precision but higher recall scores over all datasets.

Overall, these ablation studies demonstrate the comprehensive effectiveness of our method in correcting noisy labels by combining the two essential steps: recalling unlabeled labels and eliminating inaccurate labeled labels. By utilizing the knowledge from PLMs, our method can effectively address the issue of insufficient coverage of fine-grained entity types in the training samples weakly annotated by the large language model ChatGPT.

## 5.3 Detailed Analysis

**Effect of Loss Adjustment**    As previously discussed, we have developed a strategy to adjust

| Dataset | Model | P | R | F1 |
|---------|-------|---|---|-----|
| **With Distantly Annotated Training Samples** | | | | |
| OntoNotes | Baseline | 87.8±0.9 | 80.0±0.6 | **83.7±0.2** |
| | w/o unl. | **89.3±0.2** | 76.0±1.0 | 81.8±0.3 |
| | w/o ina. | 83.9±0.6 | **82.1±0.8** | 83.0±0.2 |
| WiKi | Baseline | 82.9±0.1 | 86.1±0.2 | **84.3±0.2** |
| | w/o unl. | **83.5±0.1** | 84.2±0.3 | 83.8±0.1 |
| | w/o ina. | 70.7±0.4 | **90.9±0.4** | 79.5±0.2 |
| **With ChatGPT Annotated Training Samples** | | | | |
| OntoNotes | Baseline | 88.8±0.9 | 85.4±0.9 | **87.1±0.1** |
| | w/o unl. | **90.2±0.5** | 80.1±0.6 | 84.9±0.2 |
| | w/o ina. | 85.4±0.6 | **86.1±0.2** | 85.7±0.3 |
| WiKi | Baseline | 82.2±0.5 | 83.1±0.2 | **82.6±0.4** |
| | w/o unl. | **83.0±0.4** | 79.8±0.4 | 81.3±0.1 |
| | w/o ina. | 80.5±0.2 | **83.3±0.2** | 81.8±0.1 |
| **With Crowdsourced Annotated Training Samples** | | | | |
| Ultra-Fine | Baseline | 53.2±0.7 | 48.1±0.5 | **50.5±0.2** |
| | w/o unl. | **59.5±1.2** | 41.0±0.7 | 48.5±0.3 |
| | w/o ina. | 51.5±0.6 | **48.8±0.4** | 50.1±0.1 |

Table 5: Ablation studies on three different types of weakly label.

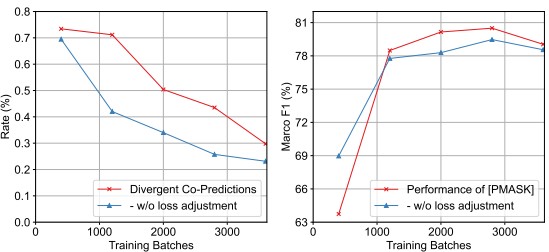

Figure 3: **Left**: The rate curves of divergent co-predictions on ChatGPT annotated training set in WiKi dataset. **Right**: The performance curves of our co-prediction model in WiKi test set, and macro F1 scores of [PMASK] are reported.

losses for labels that demonstrate divergent co-predictions. This approach aims to help the model maintain its ability to predict divergent results on noisy labels. To assess the efficacy of this strategy, we present the evolution trend of divergent co-predictions during the training process.

Figure 3 displays the evolution curve of divergent co-predictions generated on the WiKi training set (weakly labeled by ChatGPT). As the model training progresses, the number of divergent co-predictions decreases gradually. According to the *memory effect* theory, the reason may be that both [PMASK] and [NMASK] are gradually fitting more noisy labels. However, our loss adjustment strategy allows the model to generate more divergent co-predictions on the training data.

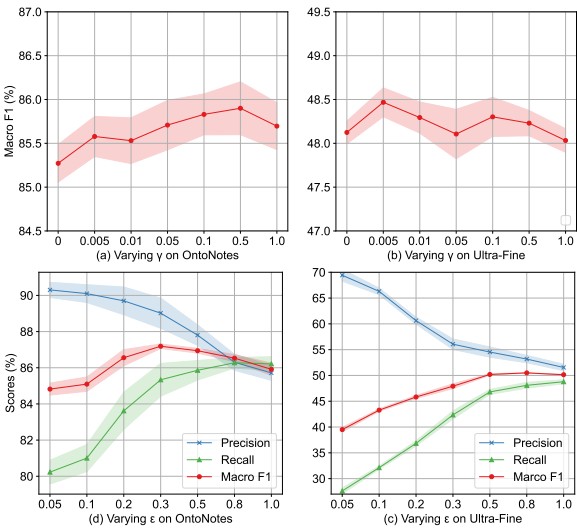

Figure 4: Performances on Ultra-Fine and OntoNotes when using different loss weight $\gamma$ and threshold $\epsilon$.

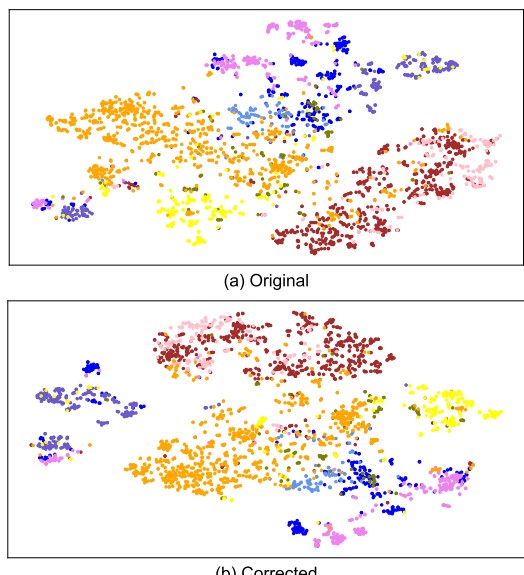

Figure 5: T-SNE virtualization of [PMASK] representations on OntoNotes test set: optimized by (a) original and (b) corrected distantly annotated training sets. The entity type "/other" is removed, as well as the types with less than 50 samples.

Training deep models on noisy labels can negatively impact their performance on unseen data. Therefore, we report the model's performance on WiKi test set. As seen, our strategy leads to a better peak generalization performance. This finding suggests that the labels with divergent co-predictions may indeed contain potentially noisy labels, and our strategy can mitigates their negative impact by restricting the model learning them.

**Sensitivity of Hyperparameters**   We explore the performance changes with respect to different $\gamma$ (Eq.5) and $\epsilon$ (Eq.7).

Selecting an appropriate $\gamma$ is crucial for the co-prediction model to effectively identify noisy labels. Since it is difficult to directly evaluate the noise detection ability, selecting an appropriate $\gamma$ relies on the model's generalization performance. Figure 4(a, b) present the results on Ultra-Fine and OntoNotes. To achieve optimal performance, hyper-parameter tuning for $\gamma$ is necessary when training the co-prediction model in real-world scenarios.

The performance analysis of corrected training sets generated with different threshold $\epsilon$ is presented in Figure 4(c, d). The results clearly demonstrate the impact of varying $\epsilon$ values on the corrected training sets. A small $\epsilon$ can improve the precision score significantly, but it may also adversely affect the recall score. Therefore, choosing a reasonable $\epsilon$ value requires striking a balance between precision and recall scores to ensure optimal model performance.

**Visualization**   To demonstrate the effectiveness of our noise correction method, we visualize [PMASK] representations optimized separately by the original and corrected training sets. We choose to display the fine-grained type since some types have only a few test samples. Upon comparing Figure 5(a) and (b), we observe more prominent margins and clearer decision boundaries between different types after noise correction. However, some clusters still contain confusing samples. We attribute this to the fact that many samples have multiple fine-grained entity types (such as "/organization/government" and "/location/country"), but only one can be reported. This also highlights the complexity of noise correction in FET, as it requires determining the correctness of each type in the label set, rather than simply selecting one correct label.

# 6   Conclusion

This paper presents a novel noise correction method for FET through the utilization of co-prediction prompt tuning. Our approach is not only straightforward but also highly effective, leveraging a pretrained language model that has been fine-tuned with a co-prediction prompt. This fine-tuned model is capable of identifying and rectifying noisy labels. To evaluate the performance of our method, we con-

ducted experiments on three publicly available FET datasets, each containing different types of training samples, including those annotated through distant supervision, ChatGPT, and crowdsourcing. The results of our experiments demonstrate the versatility of our noise correction method, as it significantly enhances the quality of the training data in all three scenarios. Consequently, it leads to a substantial improvement in the overall performance of the FET model.

## Limitations

We have developed a co-prediction prompt specifically for fine-tuning pre-trained masked language models. However, it is important to note that this is just one example of co-prediction prompt tuning, and there exist numerous possibilities for designing more engaging prompts. For instance, one can consider increasing the number of [MASK] tokens or designing prompts with more instructive words. These alternative approaches can be complemented by a novel process that utilizes the co-prediction results to further enhance the accuracy of noise correction. In addition, our method does not rely on prior knowledge or information about entity types, such as predefined hierarchical structures or detailed type introductions. While this information is often not readily available, it can be useful for identifying noisy labels.

## Ethics Statement

To address ethical concerns, we provide the two detailed description: 1) All experiments were conducted on existing datasets derived from public scientific papers or technologies. 2) Our work does not contain any personally identifiable information and does not harm anyone.

## Acknowledgements

This work was supported by Strategic Priority Research Program of Chinese Academy of Sciences (N0. XDC02040400).

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

# A Automatic Labeling with ChatGPT

In order to reduce both the cost and time required for labeling, we choose to relabel only a subset of high-quality training samples in OntoNotes and Wiki datasets.

Firstly, we apply a frequency-based filtering approach to remove entity mentions from the training samples, specifically discarding those with a frequency lower than 10 in OntoNotes and 20 in Wiki. Additionally, nonsensical non-entity mentions such as "yes" and "please" are also removed. Finally, we randomly extract 3,000 and 9,000 training samples for OntoNotes and Wiki datasets, respectively.

These samples are merged with an artificially designed prompt and then fed into *GPT-3.5 text-davinci-003* model via its official API[3] for relabeling. For parameters, we use top-p 1, with temperature 0.7. An example of the labeling process is shown in Figure 6.

# B Hyperparameters

We use a grid search to find the best hyperparameters depending on development set performances. The hyperparameters we used to fine-tuning BERT and correct noisy labels in three datasets are listed in Table 6.

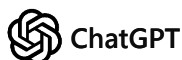

Fine-grained entity typing aims to assign semantic types to entities in texts. The type set contains: /person, / location, /organization, /other, /location/city, ......
For example:
Columbia Savings is a major holder of so - called junk bonds. The semantic types of "Columbia Savings" contain: /organization, /organization/company.
Question:
President Chen further stated that after Taiwan joins the World Trade Organization-LRB-WTO-RRB. The semantic types of "President Chen" contain:

/person, /person/political_figure

Figure 6: An example of using ChatGPT to label entity mentions.

| Parameters | OntoNotes | WiKi | Ultra-Fine |
|---|---|---|---|
| Batch size | 16 | 16 | 16 |
| Learning Rate | 3e-6 | 3e-6 | 2e-5 |
| Adam epsilon | 1e-8 | 1e-8 | 1e-8 |
| Warmup ratio | 0.0 | 0.0 | 0.0 |
| Emb. dropout | 0.2 | 0.2 | 0.2 |
| Weight decay | 0.01 | 0.01 | 0.01 |
| Clipping grad | 0.1 | 0.1 | 0.1 |
| Loss weight $\gamma$ | 0.1 | 0.1 | 0.005 |
| Threshold $\epsilon$ | 0.2 | 0.05 | 0.8 |
| $\epsilon$ on ChatGPT | 0.3 | 0.5 | - |

Table 6: Hyper-parameters in OntoNotes, WiKi and Ultra-Fine datasets.

---
[3]https://openai.com/api/
