# OpenReview forum: "Learning to Correct Noisy Labels for Fine-Grained Entity Typing via Co-Prediction Prompt Tuning"
_EMNLP/2023/Conference — EMNLP 2023 Findings_

### Official Review · Reviewer_MVhU · 2023-08-03

**Typos Grammar Style And Presentation Improvements:** 1. There are some type errors, for ex…
**Soundness:** 4

**Excitement:**

3: Ambivalent: It has merits (e.g., it reports state-of-the-art results, the idea is nice), but there are key weaknesses (e.g., it describes incremental work), and it can significantly benefit from another round of revision. However, I won't object to accepting it if my co-reviewers champion it.

**Justification For Ethical Concerns:**

N.A.

**Missing References:**

The proposed co-prediction method is related to negative learning, suggest to add some references in this domain.

**Paper Topic And Main Contributions:**

This paper focuses on noise label correction for fine-grained entity typing (FET), and proposes a label-cleaning method based on co-prediction prompt to address the problem, and utilizes LLM (such as ChatGPT) to generate weakly labeled training data for FET. Extensive experiments demonstrate the effectiveness of the proposed method.

**Questions For The Authors:**

Pls refer to reason to reject.

**Reasons To Accept:**

1. The paper is well-written and organized;
2. The proposed method is simple but effective, and may be easy to implement in real world applications, which can recall the unlabeled labels as well as eliminating inaccurate labels without estimating noise distributions, which can be used in various noise scenarios
3. This paper proposes a novel prompt-base noise correction model: (1) leverage the knowledge of PLM for noise detection and correction; (2) design two kinds of prompts with opposite objectives which provide comprehensive predictions, and highlights the effectiveness of using LLM (e.g., ChatGPT) to generate weakly labeled training data, which may be better than the tradition methods like crowdsourcing, it may be inspire how to take advantage of LLMs to handle the similar problem for NLP research community.


**Reasons To Reject:**

1. The proposed co-prediction method is related to negative learning, suggest to add some references in this domain.
2. The model reall the unlabeled labels if the prediction of [PMASK] and [NMASK] is consensus, I wonder how to handle if the prediction is divergence for some unlabeled labels in the recalling procedure.
3. The experiments are incomplete, for example, (1) The ablation study of loss adjustment, the coefficient γ is gradually decreasing during the fine-tuning process, and thus I wonder whether the performance is sensitive to the decreasing schedule, and how to obtain a proper decreasing schedule in practice? (2) The proposed method is partially sensitive to the threshold, however the analysis of ϵ is missing; (3) How to determine the parameter δ to make sure terminating the training process at the appropriate time? (4) The threshold in Eq. (7) should be small if there are many inaccurate labels, otherwise it should be relatively large when having many unlabeled data, suggest to add more experiments for further analysis.


**Reproducibility:**

4: Could mostly reproduce the results, but there may be some variation because of sample variance or minor variations in their interpretation of the protocol or method.

**Reviewer Confidence:**

4: Quite sure. I tried to check the important points carefully. It's unlikely, though conceivable, that I missed something that should affect my ratings.

---

> ### Author Rebuttal · Authors · 2023-08-28
>
> Thank you sincerely for providing your valuable comments. We genuinely appreciate your feedback, and we would like to respond to each point with utmost sincerity and kindness:
>
> Q1: About negative learning.
>
>
> A1: We appreciate the reviewer's point regarding negative learning. We will carefully consider the relevance of negative learning to our method and further improve the related work section accordingly.
>
>
> Q2：About the recalling procedure of the prediction is divergence for some unlabeled labels.
>
>
> A2: If the prediction is divergence of [PMASK] and [NMASK] for a certain unlabeled label, it is important to consider whether this label is correct. Therefore, as described in the section "Eliminating Inaccurate Labels," we calculate the divergence score (Eq.7) and compare it with a threshold value, \epsilon . If the divergence score is greater than \epsilon , we consider this recalled label from the model's prediction to be potentially incorrect. If it is less than \epsilon, we consider the predicted label to be correct. The choice of threshold value \epsilon is flexible, and we discuss its sensitivity in the "Sensitivity of Hyperparameters" section.
>
> Q3：About the  ablation study of loss adjustment.
>
>
> A3: We will add additional experimental data on using different decreasing schedules of \gamma.
>
>
> Q4: About the analysis of \epsilon.
>
>
> A4: In Figure 4, we present the impact of different values of \epsilon  and draw the following conclusion: A small \epsilon can significantly enhance the precision score; however, it may have a negative impact on the recall score. We appreciate the thoroughness of the reviewer and, as a result, we intend to provide further analysis on the parameter \epsilon in the "Sensitivity of Hyperparameters" section.
>
> Q5: About how to determine the parameter \gamma to make sure terminating the training process at the appropriate time.
>
>
> A5: During the co-prediction prompt tuning process, the model stops training once it reaches early stopping criteria, which is typically determined by achieving the best generalization performance on the development set. The parameter \gamma primarily affects the fitting differences of [PMASK] and [NMASK] to the noise labels. By adjusting the value of \gamma, we can control the degree of alignment between the prediction results of [PMASK] and [NMASK]  on the noise labels, thereby influencing the model's ability to effectively correct the noise in the dataset.
>
>
>
> We genuinely appreciate your feedback and suggestions, and we will incorporate them into our work to improve its quality. We highly appreciate your time, effort, and consideration.

---

### Official Review · Reviewer_yPJ4 · 2023-08-05

**Soundness:** 3

**Excitement:**

3: Ambivalent: It has merits (e.g., it reports state-of-the-art results, the idea is nice), but there are key weaknesses (e.g., it describes incremental work), and it can significantly benefit from another round of revision. However, I won't object to accepting it if my co-reviewers champion it.

**Paper Topic And Main Contributions:**

This paper targets the fine-grained entity typing task which classifies the mention of an entity given the context information. Different from typical classification approaches, this paper formulate entity typing as a masked token prediction task. Firstly, the authors compose a textual input which includes the context information, the mention of the entity and the typing information (masked tokens); Secondly, the authors use a pre-trained language model to predict the masked token; Thirdly, the predicted masked token as well as the probabilities are used to infer the entity type.

**Reasons To Accept:**

1. The proposed "co-predict" mechanism is interesting which includes both positive and negative label in one prompt.

2. The proposed model outperforms some existing works on public data sets.


**Reasons To Reject:**

1. Since the performance of the proposed model heavily rely on the capability of the pretrained language model, a SOTA backbone network can be adopted rather than the BERT-base. For example, the author can use LLaMa or ChatGLM as the backbone pre-train model.


2. The proposed "co-predict" model is similar to contrastive learning, the author needs to include contrastive learning-based approaches as baselines in the experiment section.

3. Since the major contribution of this paper is the "co-predict" mechanism, the authors need to evaluate the performance "co-predict" in detail. For example, the authors can modify the prompt without the "rather than" part as an baseline.


**Reproducibility:**

3: Could reproduce the results with some difficulty. The settings of parameters are underspecified or subjectively determined; the training/evaluation data are not widely available.

**Reviewer Confidence:**

3: Pretty sure, but there's a chance I missed something. Although I have a good feel for this area in general, I did not carefully check the paper's details, e.g., the math, experimental design, or novelty.

---

> ### Author Rebuttal · Authors · 2023-08-28
>
> Thank you sincerely for providing your valuable comments. We genuinely appreciate your feedback, and we would like to respond to each question as follows:
>
> Q1: About a SOTA backbone network can be adopted rather than the BERT-base, such as LLaMa or ChatGLM.
>
>
> A1: Our research focuses on addressing the issue of noisy labels in datasets obtained through automatic annotation techniques. In our noise correction method, we fine-tune BERT using co-prediction prompts to generate multiple prediction results. These prediction results are then utilized to identify and correct the noisy labels. By utilizing a SOTA backbone, such as LLaMa or ChatGLM, our method may potentially achieve more reliable prediction results, thereby improving the accuracy of noise label detection and correction. This, in turn, enhances the quality of datasets obtained through automatic annotation techniques.
>
>
> Q2：About the proposed "co-predict" model is similar to contrastive learning.
>
>
> A2：It is important to note that our proposed "co-predict" model may have inherent differences from "contrastive learning." Contrastive learning aims to learn meaningful representations by maximizing similarity between similar instances and minimizing similarity between dissimilar instances. But, our "co-prediction prompt tuning" is designed to guide BERT in generating multiple prediction results that target diverse predictive abilities. Therefore, their motivations are different. Furthermore, in the field of fine-grained entity typing, there have been no prior works that specifically employ contrastive learning for noise correction. Therefore, it is challenging to find corresponding studies as baselines for comparison. We hope the reviewer understands and forgives this limitation.
>
>
> Q3：About modifying the prompt without the "rather than" part as an baseline.
>
>
> A3: During the noise correction process, the prediction results generated by co-prediction prompts form the core of recalling unlabeled labels and eliminating inaccurate labels. Therefore, we can not remove the "rather than" part as a baseline to achieve noise correction independently.
> In evaluating of the quality of our method's corrected dataset, as mentioned in the "Experimental Details" section, we fine-tune vanilla BERT using the corrected dataset based on standard prompts (without "rather than [NMASK]"). Additionally, we have fine-tuned vanilla BERT on the corrected dataset by using our co-prediction prompts as a baseline.
>
>
> We genuinely hope that our responses address the queries and contribute to a positive outcome. We highly appreciate your time, effort, and consideration.

---

### Official Review · Reviewer_FZzX · 2023-08-12

**Soundness:** 3

**Excitement:**

3: Ambivalent: It has merits (e.g., it reports state-of-the-art results, the idea is nice), but there are key weaknesses (e.g., it describes incremental work), and it can significantly benefit from another round of revision. However, I won't object to accepting it if my co-reviewers champion it.

**Paper Topic And Main Contributions:**

The paper proposed a co-prediction prompt tuning approach to correct incorrect labels in entity typing data. The proposed approach achieves improvement over vanilla prompt tuning. Improvements are shown on three types of noisy training data, with the most significant improvement seen on distantly annotated data.

**Questions For The Authors:**

A. In Table 7 of the MLMET paper [2], a vanilla BERT-base model is able to achieve 75.9 Micro-F1 on Ontonotes, which is significantly higher than the reported 71.2 in the paper. Is there any reason that might have caused the discrepency?

B. Details of decreasing \gamma during the training process are not mentioned.

C. Some details in the “verbalizer selection” section are not clear (e.g. how do you compute the probability for "/organization/company/news” which consists of multiple words and each word might consist of multiple tokens?)

D. How is the “/other” category in Ontonotes handled during label correction?

[2] Ultra-Fine Entity Typing with Weak Supervision from a Masked Language Model. ACL2021

**Reasons To Accept:**

- The paper is overall well written and easy to follow. The high-level intuition of the method is clear. The method is novel and seems to be a general method that can be applied to other classification tasks.
- The proposed approach achieve improvements on different types of annotated data (distantly/ChatGPT/human).
- The authors experiment with three datasets and carry out detailed analysis.

**Reasons To Reject:**

- While experimental results shows clear improvements in terms of end task performance (results of correction + finetuning), no evaluations (either by case study or quantative experiments) on the corrected labels themselves (results of correction only) are conducted. It is also unclear why a BERT-base model is able to correct labels annoated by human. Additionally, the training set and test set of Ultra-fine are sampled from the same distribution and the results in Table 4 indicates that altering the training set leads to improvement. This result is unintuitive and requires further explanation.
- While the authors claim that the proposed approach outperform previous SOTA, the results on OntoNotes are siginificantly lower than some of the recent methods [1] (76.3 vs 81.4). MLMET is used as a baseline for Ultra-fine but its results on Ontonotes are not reported (80.35).

[1] Ultra-fine entity typing with indirect supervision from natural language inference. TACL2022

**Reproducibility:**

4: Could mostly reproduce the results, but there may be some variation because of sample variance or minor variations in their interpretation of the protocol or method.

**Reviewer Confidence:**

4: Quite sure. I tried to check the important points carefully. It's unlikely, though conceivable, that I missed something that should affect my ratings.

---

> ### Author Rebuttal · Authors · 2023-08-28
>
> Thank you sincerely for providing your valuable comments. We would like to respond to each question as follows:
>
> Q1:About why a BERT-base model is capable of correcting labels annotated by humans (e.g., Ultra-fine dataset, [1] Ultra-fine entity typing. 2018 ACL).
>
>
> A1: The Ultra-fine dataset, specifically the Ultra-fine entity typing dataset from the 2018 ACL [1], consists of annotations provided by five crowd workers on Mechanical Turk. This dataset includes a total of 2519 entity types. Due to the limited domain expertise of the crowd annotators, accurately labeling all 2519 types can be challenging, often resulting in missing annotations. In our noise correction method, we start by  fine-tuning a pre-trained masked language model, such as BERT, using a co-prediction prompt. Throughout the training process, we devise an optimization objective that allows multiple prompts (e.g., [PMASK] and [NMASK]) to adapt to varying degrees of noise. And the co-prediction prompt is designed to guide the BERT to generate multiple prediction results that target diverse predictive abilities. This training approach enables the BERT model to identify and correct noisy labels based on the co-prediction results.  For Ultra-fine dataset, these different prediction results can be combined to recover unlabeled labels and address the issue of missing annotations.
>
>
> Q2: About the results on OntoNotes are lower than the recent method ([2] Ultra-fine entity typing with indirect supervision from natural language inference. 2022 TACL)
>
>
> A2: It is important to note that the OntoNotes dataset has two versions: the original version and the augmented version introduced by the paper [1] (Ultra-fine entity typing. 2018 ACL). The experimental results presented in paper [2] were conducted on the augmented version, while our experiments were exclusively conducted on the original dataset. To ensure a fair comparison, we did not take the method presented by paper [2] as our baseline. In next version, we will retest the method presented by paper [2] on the original OntoNotes dataset and present the comparative results in the paper.
>
>
> Q3:MLMET ([3] Ultra-Fine Entity Typing with Weak Supervision from a Masked Language Model. 2021 ACL) is used as a baseline for Ultra-fine but its results on Ontonotes are not reported (80.35).
>
>
> A3: Similarly, MLMET conducted experiments on the augmented version of the OntoNotes dataset, while we conducted experiments on the original dataset. So,we did not reported the results of MLMET on OntoNotes dataset.  Additionally, the performance drops can be attributed to the variations in training the vanilla BERT-base model on different versions of the OntoNotes dataset. This is the primary reason for the decrease in performance, rather than any other unidentified factors.
>
>
>
>
> Q4: About the details of decreasing γ during the training process.
>
>
> A4: The parameter \gamma is gradually decreased during the training process until it reaches the marginal value \delta. This process occurs in the first training epoch, and \gamma decreases linearly.  For instance, if an epoch consists of 1000 batches and \delta is set to 0.1, then \gamma = 1 - 0.0009 * (number of batches).  We genuinely appreciate your meticulous, and we will add these details in the next version of our paper.
>
>
> Q5: About the details in the "verbalizer selection" section.
>
>
> A5:  As mentioned in our paper, we utilize the average embedding of type tokens to initialize soft words. For example, when decomposing "/organization/company/news," we obtain three individual words: "organization," "company," and "news." These word embeddings can be acquired from the embedding layer of the BERT model by using their corresponding IDs in the BERT tokenizer's vocabulary. By summing and averaging the embeddings of "organization," "company," and "news," we derive the representation for the type "/organization/company/news."
>
>
> Q6: About how the "/other" category in OntoNotes is handled during label correction.
>
>
> A6: In our method, we capture noisy patterns within the noisy dataset through co-prediction prompt tuning BERT. Subsequently, for each instance of entity classification, we propose a label correction process that involves recalling unlabeled labels and eliminating inaccurate ones. When it comes to the "/other" category, we do not treat it differently. If "/other" is a missing label for an instance of entity typing, it would be corrected through the process of recalling unlabeled labels. Conversely, if "/other" is an incorrectly assigned label, it may be rectified through the process of eliminating inaccurate labels.
>
>
> We genuinely hope that our responses address the queries and contribute to a positive outcome. We highly appreciate your time, effort and consideration.

---

### Meta-Review · Area_Chair_swXq · 2023-09-17

**Recommendation:** 2

**Metareview:**

The paper proposed a co-prediction prompt tuning method for the entity typing task. The method could correct the noisy labels through masked token prediction. The experimental results on three datasets showed that the proposed model mostly outperformed existing work.

Strength: The proposed "co-predict" mechanism is interesting and somewhat novel. The whole paper is well-organized and clear.

Weakness: 1) Actually, BERT is also based on the masked token prediction. So it is not very clear whether BERT can actually correct human-annotated labels, which should be investigated further. 2) co-prediction prompt is similar to contrastive learning. The authors should explain their differences through illustration or experimental comparisons. 3) More experimental analyses/discussions are needed.

---

### Decision · Program_Chairs · 2023-10-07

**Decision:**

Accept-Findings

**Comment:**

The paper proposed a co-prediction prompt tuning method for the entity typing task. The method could correct the noisy labels through masked token prediction. The experimental results on three datasets showed that the proposed model mostly outperformed existing work.

Strength: The proposed "co-predict" mechanism is interesting and somewhat novel. The whole paper is well-organized and clear.

Weakness: 1) Actually, BERT is also based on the masked token prediction. So it is not very clear whether BERT can actually correct human-annotated labels, which should be investigated further. 2) co-prediction prompt is similar to contrastive learning. The authors should explain their differences through illustration or experimental comparisons. 3) More experimental analyses/discussions are needed.